# Studying the context of psychoses to improve outcomes in Ethiopia (SCOPE): Protocol paper

**Charlotte Hanlon**[1,2,3]*, **Tessa Roberts**[1,4], **Eleni Misganaw**[5], **Ashok Malla**[6], **Alex Cohen**[7], **Teshome Shibre**[8], **Wubalem Fekadu**[2], **Solomon Teferra**[2], **Derege Kebede**[9], **Adiyam Mulushoa**[2], **Zerihun Girma**[2], **Mekonnen Tsehay**[2], **Dessalegn Kiross**[10], **Crick Lund**[1,11], **Abebaw Fekadu**[2,3], **Craig Morgan**[4], **Atalay Alem**[2]

1 Health Service and Population Research Department, Institute of Psychiatry, Psychology and Neuroscience, Centre for Global Mental Health, King's College London, London, United Kingdom, 2 Department of Psychiatry and WHO Collaborating Centre in Mental Health Research and Capacity Building, School of Medicine, College of Health Sciences, Addis Ababa University, Addis Ababa, Ethiopia, 3 Centre for Innovative Drug Development and Therapeutic Trials for Africa (CDT-Africa), College of Health Sciences, Addis Ababa University, Addis Ababa, Ethiopia, 4 ESRC Centre for Society & Mental Health, King's College London, London, United Kingdom, 5 Mental Health Service User Association, Addis Ababa, Ethiopia, 6 Department of Psychiatry and Douglas Mental Health Institute, McGill University, Montreal, Canada, 7 Department of Population Health, London School of Hygiene and Tropical Medicine, London, United Kingdom, 8 Horizon Health Network Zone 3, New Brunswick, Canada, 9 Department of Preventive Medicine, School of Public Health, School of Medicine, College of Health Sciences, Addis Ababa University, Addis Ababa, Ethiopia, 10 Victoria University of Wellington, School of Nursing, Midwifery and Health Practice, Wellington, New Zealand, 11 Department of Psychiatry and Mental Health, Alan J Flisher Centre for Public Mental Health, University of Cape Town, Cape Town, South Africa

* charlotte.hanlon@kcl.ac.uk

**Data Availability Statement:** Deidentified research data will be made available when the study is completed and published.

## Abstract

### Background

Global evidence on psychosis is dominated by studies conducted in Western, high-income countries. The objectives of the Study of Context Of Psychoses to improve outcomes in Ethiopia (SCOPE) are (1) to generate rigorous evidence of psychosis experience, epidemiology and impacts in Ethiopia that will illuminate aetiological understanding and (2) inform development and testing of interventions for earlier identification and improved first contact care that are scalable, inclusive of difficult-to-reach populations and optimise recovery.

### Methods

The setting is sub-cities of Addis Ababa and rural districts in south-central Ethiopia covering 1.1 million people and including rural, urban and homeless populations. SCOPE comprises (1) formative work to understand care pathways and community resources (resource mapping); examine family context and communication (ethnography); develop valid measures of family communication and personal recovery; and establish platforms for community engagement and involvement of people with lived experience; (2a) a population-based incidence study, (2b) a case-control study and (2c) a cohort study with 12 months follow-up involving 440 people with psychosis (390 rural/Addis Ababa; 50 who are homeless), 390 relatives and 390 controls. We will test hypotheses about incidence rates in rural vs. urban populations and men vs. women; potential aetiological role of khat (a commonly chewed

**Funding:** This study is funded by Wellcome Trust (https://wellcome.org/) grant 222154/Z20/Z awarded to CH (PI), AA, AF, EM, ST, TR, CM, CL, SM, AC. The funder and the sponsor (KCL) did not play any role in the study design, data collection and analysis, decision to publish, or preparation of the manuscript.

**Competing interests:** The authors have declared that no competing interests exist.

plant with amphetamine-like properties) and traumatic exposures in psychosis; determine profiles of needs at first contact and predictors of outcome; (3) participatory workshops to develop programme theory and inform co-development of interventions, and (4) evaluation of the impact of early identification strategies on engagement with care (interrupted time series study). Findings will inform development of (5) a protocol for (5a) a feasibility cluster randomised controlled trial of interventions for people with recent-onset psychosis in rural settings and (5b) two uncontrolled pilot studies to test acceptability, feasibility of co-developed interventions in urban and homeless populations.

## Introduction

Psychoses, such as schizophrenia, affect more than 23 million people worldwide, contribute substantially to the global burden of disease and are associated with high rates of disability and mortality [1], particularly in low resource settings where most never receive treatment [2]. The onset of psychosis is mostly in late adolescence and early adulthood, increasing the salience to low- and middle-income countries (LMICs) with younger population profiles. Nonetheless, although over 85% of the world's population lives in Asia, Africa, Latin America, and the Caribbean, less than 10% of psychosis research is carried out in these settings [3]. Consequently, our knowledge of the early stages of psychoses, especially of the basic epidemiology, risk factors, and early course and outcome, is based almost entirely on research from North America, Western Europe, and Australia. Recent evidence indicates that psychoses are highly heterogeneous in their distribution, aetiology, incidence, presentation and outcome [3]. Population-based research in more diverse contexts on incidence, and predictors of onset and outcomes among representative samples of people with psychosis is critical to develop appropriate evidence-based interventions [4].

There has been relative neglect of psychosis in global mental health research [5], with very few methodologically robust population-based studies of psychosis in LMICs [3]. Most existing studies are based either on clinical samples, likely to be highly unrepresentative of the wider population as only a small proportion of people with psychosis access mental health services [6,7], or on those with long duration of illness in community settings [8]. The ongoing INTREPID II programme in India, Nigeria and Trinidad [9], based on extensive pilot work [10–12], provides methodology that is feasible to implement in diverse LMICs, thus enabling cross-country comparisons. To date INTREPID methods have not been applied in low-income or Eastern African countries. This is an important gap, given the great diversity of health systems, health profiles, burdens of disease, and social, economic and cultural factors across Africa.

In Ethiopia, SCOPE will align with INTREPID epidemiological methods and build on two internationally-recognised community-based studies of people with psychosis: the Butajira course and outcome study ('Butajira study') [13,14] and the PRIME (Programme for Improving Mental health carE) study [15,16]. In the Butajira study, an epidemiological sample of 359 people with clinician-confirmed diagnoses of schizophrenia was recruited through community case-finding methods and followed up for an average of 10 years [13,14]. Important findings from this study were: a high treatment gap (90% lifetime), long duration of untreated psychosis (DUP; mean 7.6 years), 7% street homelessness at baseline [8,17,18]; low rates of complete remission [13], high suicide attempt rates [19] and higher levels of disability [20,21], perpetrated violence and violent victimisation [22] and premature mortality (27.7 years of life lost; standardised mortality ratio 302.7) [23] compared with the general population.

Transgenerational transmission of disadvantage was shown [24], with high levels of caregiver burden [25], economic impact [26] and experience of stigma [27]. In the PRIME study, 300 people with severe mental illness (85.3% with primary psychosis) were detected in the community [28,29]. Key findings were lifetime and current access gaps for biomedical care of 41.8% and 59.9%, respectively, with corresponding figures for faith and traditional healing of 15.1% and 45.2% [7]; only 11.3% received minimally adequate biomedical care in current episode [7]; long DUP (median 5 years); high rates of restraint (25.3% in the preceding 12 months); high exposure to traumatic events [7,30], high lifetime experience of homelessness (36.3%) [7], higher discrimination in urban residents [31]; higher poverty [32,33] and food insecurity [34] compared to population controls. Both the Butajira and PRIME studies were limited by potential prevalence bias and limited recruitment from religious healing sites.

The aetiology of psychosis is multifactorial, contributed to by social and environmental risk factors alongside genetic and developmental risks [35]. In the Butajira and PRIME studies, potentially salient psychosocial risk factors and outcomes for the Ethiopian context were not examined. The evidence gaps about potential risk factors (khat use, trauma exposure), targets for intervention (family communication) and outcomes (personal recovery) will now be described.

i. Khat use: Khat leaves contain *Catha Edulis*, an amphetamine-like substance [36]. Khat is widely used across the Horn of Africa and its diaspora [37,38]. In Ethiopia, use is increasing, with 15.8% nationally reporting current use [39], but over 50% of adults in some districts [40] and high levels in students [41]. In a case-control study from Somalia, age of onset of khat use was associated with clinical ratings of current psychotic symptoms [42], indicating a potential role as a risk factor for onset of primary psychosis. However, the relationship between culturally relevant patterns of khat use [43] and incidence and early course of psychoses has not been investigated.

ii. Trauma: In studies conducted in high-income countries, co-morbidity of post-traumatic stress disorder (PTSD) in people with psychosis ranges from 13–55% [44]. Exposure to traumatic events is a risk factor for developing psychosis [45] and associated with poorer outcomes [46]. In LMICs, traumatic experiences are prevalent among the general population, with an estimated 8% affected by PTSD [47]. In people with psychosis, traumatic experiences are likely to be more common, arising from restraint or coercive treatments [7], accidents or sexual assault [7] or other forms of violent victimisation [22]. The limited evidence available about the role of traumatic exposures in onset and course of psychosis in LMICs indicates variation of the association across setting [48].

iii. Family communication and involvement: High 'expressed emotions' from family members, specifically hostility, critical comments and over-involvement, are associated with increased risk of relapse and hospitalisation of people with psychosis [49]. However, a recent meta-analysis [50] identified only one study of expressed emotion from Africa [51]. Important cultural variability of expressed emotion is recognised [50,52]. Given the crucial role of the family in caregiving in Ethiopia and other African countries [53], local evidence is urgently needed on relevant family communication patterns to inform intervention.

iv. Personal recovery: Defined as *"a way of living a satisfying, hopeful and contributing life even within the limitations caused by illness" [54],* personal recovery is increasingly recognised as an essential goal for intervention [55]. However, the need to explore applicability of this concept in non-Western populations has been noted [56]. A study from Ethiopia explored the concept of recovery in an urban, hospital-based study [57]. However, evidence from a

representative sample is required to ensure innovations are based on priorities and values of people with psychosis.

In high-income countries, "early intervention for psychosis" (EIP) models, comprising packages of treatments for people with psychosis at first contact with services, are effective and cost-effective in those settings [58,59]. There have been a very small number of studies from middle-income countries where intensive specialist-led EIP models were adapted [60–63], but we are not aware of any wider scale-up of these efforts. In any case, these models have been critiqued for not including efforts to achieve earlier interventions, within a 'critical period' of 2–3 years from psychosis onset, in order to mitigate the impacts of untreated psychosis [64,65]. There have been only a small number of evaluations of interventions to reduce DUP in high-income countries, with limited evidence of sustainable impacts [66], except for a high quality study testing a multi-component intervention in Norway [67]. We are not aware of studies from LMIC where impact of interventions on DUP has been evaluated. In LMICs, where DUP is substantially longer [68] and where delayed access to care has been shown to be associated with poorer functional outcomes [69], interventions to reduce DUP assume even greater importance. It has been argued that early intervention models in LMICs should look very different to their high-income country counterparts, focused on a public mental health approach [70].

Efforts to expand access to care in LMICs have included training community members in proactive case identification and linkage with care in rural populations [29,71], which have the potential to reduce DUP. However, there has been no work on interventions targeting early manifestations of psychosis or application to urban settings or homeless populations. Furthermore, evidence is needed on what interventions are required at first contact with services to optimise outcomes. Our work in Ethiopia has shown that integrating mental health care into primary healthcare in rural settings can expand access to care for people with psychosis [29], that this model is as effective as psychiatric nurse-led care [72] and delivers benefits in terms of functioning, food security [73] reduced suicidality and substantially reduced experience of discrimination and restraint [15]. However, there was minimal impact on symptom severity, 10.6% still reported experience of restraint after one year, and mortality was high. High levels of exposure to traumatic events [30] and undernutrition (23.2% underweight) were also unaddressed [74]. Furthermore, only 30% received 'minimally adequate treatment' over 12 months follow-up [15], driven by poverty, the lack of outreach and expectation of cure [75]. We have no evidence of interventions to optimise earlier interventions in urban or homeless populations. In Ethiopia, people who are homeless and have psychosis are trapped in a vicious cycle of poor health due to systematic exclusion from health care [76]. To optimise recovery of people with psychosis in Ethiopia and other LMICs, there is a need to achieve earlier first contact with care and more effective early interventions suited to the sociocultural and economic settings, that address contextually relevant needs for rural, urban and homeless populations.

In SCOPE we aim to pioneer a radical rethinking of early identification and intervention models for people with psychosis at first contact in Ethiopia, grounded in a detailed understanding of contextual needs, inclusive of difficult-to-reach populations, and based on the priorities of people with psychosis.

The specific aims of SCOPE are to:

1. Map community resources, understand help-seeking contexts, and explore concepts of family communication and involvement and personal recovery to develop contextually appropriate measures.

2. Characterise the epidemiology of psychosis in Ethiopia:

    a. Determine incidence, needs and presentation of psychosis at first contact;

b. Investigate the role of the urban environment, poverty, khat use, and traumatic experiences during the life course on onset of psychosis;

c. Identify how these exposures and additional factors, including family communication and involvement, impact on personal recovery and outcomes valued by people with psychosis over a 12-month period.

3. Based on evidence from (1) and (2), co-develop contextually grounded interventions for people with psychosis in rural, urban and homeless populations to achieve earlier and better care at first contact;

4. Assess impact of innovative identification strategies on service engagement.

5. Findings from Aims 1 to 4 will inform development of a full protocol for studies that will aim to evaluate the (5a) feasibility of trial methods and (5b) the acceptability and feasibility of co-developed interventions in rural, urban and homeless populations.

## Materials and methods

Ethical approval for SCOPE has been obtained by the Institutional Review Board of the College of Health Sciences, Addis Ababa University (Ref. 001/22/Psy; 12[th] January 2022) and the Research Ethics Committee of King's College London (Ref. HR/DP-21/22-26183; 5[th] April 2022).

The study designs to address the aims that will be discussed in this paper include (i) ethnography, resource mapping, piloting and validation studies for novel measures, and theory of change workshops (**Aim 1**); (ii) a population-based incidence study (**Aim 2a**), case-control study (**Aim 2b**) and cohort study with 12 months follow-up (**Aim 2c**) and nested qualitative study (**Aim 2c**), (iii) co-development of contextually-informed innovations (**Aim 3**), (iv) interrupted time series study (**Aim 4**). Protocols for (v) a feasibility cluster randomised controlled trial (RCT) (**Aim 5a**) and (vi) two uncontrolled pilot studies (**Aim 5b**) will be published separately. See Fig 1.

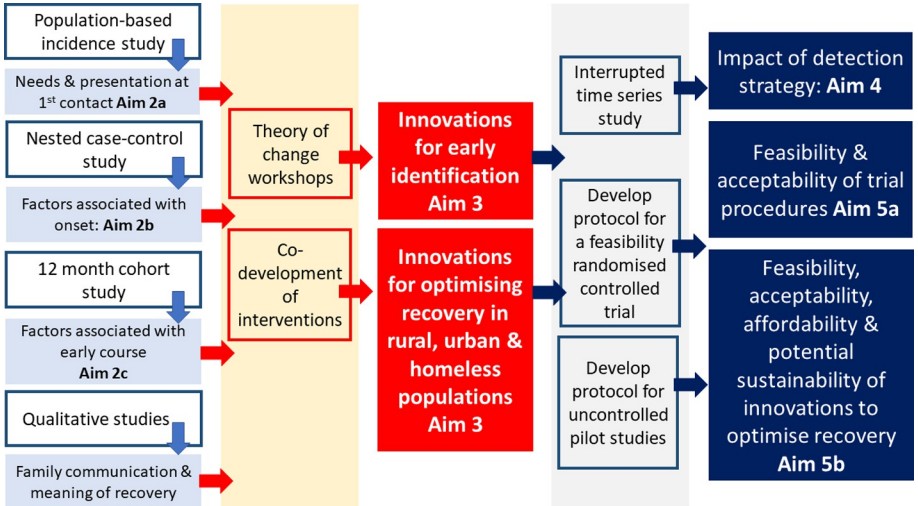

**Fig 1. SCOPE study components in relation to aims.**

## Setting

The studies will be conducted in two sites: (1) contiguous, predominantly rural districts in south-central Ethiopia (Gurage zone of Southern Nations, Nationalities and People's (SNNP) Region: Misrak Meskan, Merab Meskan, Sodo and South Sodo; Oromia region: Sodo Daci and Kersana Malima; Special district: Sabata Hawas) with an estimated total population of 713,123 people [77]; and (2) Lideta and Kirkos sub-cities of Addis Ababa, the capital of Ethiopia, with an estimated total population of 416,389 in 2016 [78]. These settings are described in detail in S1 File.

**Aim 1: Formative studies and community engagement.** The formative work for SCOPE is described briefly below. More detail is provided in S2 File.

**Resource mapping.** Research question: what are the care options, pathways and community resources to support recovery of people with psychosis?

Previous work from Ethiopia identified community resources with potential to be mobilised for people with mental health conditions in rural districts [79]. To map community resources in Addis Ababa, we are using similar methods, including collecting input from the SCOPE community advisory boards, publicly available information, and direct observations. Geographic Information Systems (GPS) coordinates for each potential resource are being recorded using Google Maps [80].

We adapted the PRIME situation analysis tool (originally developed by study co-author) to document community and health system characteristics across all study districts, complemented by desk reviews, consultations with key informants and the advisory boards [81].

**Ethnographic study.** Research question: what are the culturally important aspects of family communication and involvement in relation to people with psychosis in Ethiopia?

Ethnographic observations in 12–20 households of people with psychosis are combined with 20–30 in-depth interviews with a range of stakeholders (people with psychosis, caregivers, mental health providers, community leaders). These will investigate patterns of family interaction, impacts of mental ill-health and the status of the individual with psychosis in the family. Families will be purposively selected based on urban/rural location, trajectory of illness, and educational level of household head, recruited from the Butajira psychiatric clinic or Sodo district mental health care services (rural site), or Lideta sub-city health centres or mental health services (Addis Ababa). The person with psychosis will be required to provide informed consent and all members of the household should agree to the observation. The household head will also provide informed consent. A researcher will spend two hours at a time with each family, scheduled for different times of the day, to observe family members' activities on arrival and their interactions with the person with psychosis. Each household with a person with psychosis will be observed for an estimated 30 to 40 hours over a period of six months. The observations will be conducted by master's level research assistants, one male and one female depending on the gender of the person with psychosis. If a person with psychosis becomes unwell and requires mental health care, the researcher will liaise with a senior mental health professional in the team and advise the family to support the person to access mental health care. Field notes from participant observations will be analysed using an interpretative phenomenological approach [82] while interview transcripts will undergo thematic analysis [83] using NVIVO-12 software [84].

We will triangulate findings from both data sources.

**Instrument adaptation and development study.** Research question: what is the semantic, content, construct and convergent validity of newly developed measures of (a) family communication and involvement, and (b) personal recovery?

For each construct, we will develop, pilot and validate measures using expert consensus meetings, cognitive interviewing, a pilot study and a validation study. Existing measures were first reviewed in light of local qualitative evidence (described above for family communication and involvement, personal recovery evidence based on analysis of data from RISE trial [85]) to produce a list of potentially relevant items. This will be followed by cognitive interviewing with people with psychosis and caregivers to test comprehensibility and acceptability and inform development of a first version of the measures. This will be piloted in 200 people with psychosis and caregivers to test the psychometric properties. After adaptation based on the pilot, the final version will be validated with 400 people with psychosis and 400 caregivers to explore construct validity (using confirmatory factor analysis) and convergent validity with symptom severity and functional impairment. As there is no definitive way to determine sample size for scale validation, depending in part on how well the items are related to the construct under investigation [86], we will follow the recommended rule of thumb to use a participant-to-item ratio of 10:1 [87], assuming that we will have no more than 40 items per scale. Consensus meetings will be held with both Ethiopian and international experts, with academic, clinical, and lived experience, to review study findings at each stage. The resulting measures will be used in the epidemiological study.

## Community engagement and involvement of people with lived experience

We have established multi-sectoral advisory boards (including people with lived experience, family members, community leaders, traditional and religious healers, police, health care administrators, managers and practitioners, non-governmental sector and non-health government representatives e.g. youth, social care, women's affairs) in the Oromia and SNNP region rural sites (n = 2) and Addis Ababa (n = 1) who will provide local oversight and project ownership, support community mobilisation, and enabling trouble-shooting. Project progress and preliminary findings will be presented to the advisory boards on a regular basis to obtain feedback and contribute to the co-production of innovations to be tested in SCOPE.

The SCOPE study has benefitted from input from people with psychosis from the beginning of protocol development. Ongoing involvement is supported through (1) a co-investigator from Ethiopia with lived experience, (2) empowerment activities to support active involvement of people with psychosis from rural districts, (3) supporting the grass-roots service user association in Sodo through enabling meetings and developing peer support, (4) including lived experience feedback as a standing item at advisory board meetings, and (5) close engagement with the Mental Health Service User Association of Ethiopia and Sodo service users in research processes, including planning safeguarding procedures, reviewing information leaflets and case vignettes, instrument adaptation, and co-developing community identification strategies and rights-based interventions.

## Aim 2: Epidemiological study

Methods for the epidemiological study are closely aligned with the INTREPID study [9,10].

## Hypotheses

(2a) Incidence study

i.  Incidence of psychosis will be higher in men [88] and in urban residents [89].

(2b) Case-control study

i. Earlier age of khat initiation and/or problematic khat use [42], exposure to traumatic events prior to psychosis onset and childhood adversity [45] will be associated with increased odds of psychosis

(2c) Cohort study

i. Maladaptive family interactions [50], exposure to traumatic events [46,90], physical co-morbidity, and poverty will be associated with poorer 12-month clinical, functional and personal recovery, after adjusting for confounders.

**Sample.** We will recruit people with untreated psychosis (rural n = 240; urban n = 150), relatives (rural n = 240; urban n = 150) and age- and gender-matched population controls (rural n = 240; urban n = 150). Eligibility criteria will be identical to those detailed in the published INTREPID protocol [9] with an additional requirement that onset of psychosis was within the past two years, and extending the lower age to 15 years and removing an upper age limit (see Table 1). The reason for lowering the age for inclusion is because many cases of psychosis start in late adolescence [91]. We will seek informed consent from those aged under 18 years. If they are an emancipated minor, we will not seek permission from a guardian/responsible family member. If they are aged 15–17 years and not an emancipated minor, we will seek permission from that person before including in the study.

In addition, in the Addis Ababa sub-cities we will identify people with untreated psychosis who are homeless. Our operationalisation of the concept of homelessness was informed by stakeholders in the community advisory board. Homelessness is thus defined here as spending the night unsheltered or in other places not intended for habitation (e.g. under bridges). It includes people who can sporadically pay for shelter but excludes people who spend their days on the streets, for example, to beg, but who have stable night-time accommodation. The methodology used for the homeless sample will necessarily differ from the main epidemiological study (as outlined below) but will allow quantification of the proportion of people with untreated psychosis in Addis Ababa who are homeless. On the basis of our previous work, we expect 31 homeless people to have untreated psychosis on initial case-finding (cross-sectionally) with further incident cases recruited over time [92]. See Fig 2 for samples in the different components of the epidemiological study.

**Table 1. Eligibility criteria for SCOPE epidemiological study.**

| Inclusion criteria | Exclusion criteria |
|---|---|
| **Cases** | |
| • Aged 15 years or above<br>• Resident in the catchment area<br>• ICD-11 primary psychotic disorder<br>• Not treated with antipsychotic medication for ≥ 1 continuous month<br>• Onset within 2 years | • Moderate or severe neurodevelopmental disability (e.g. intellectual disability)<br>• Clinically manifest organic cerebral disorder<br>• Transient psychotic symptoms resulting from acute intoxication |
| **Matched controls** | |
| • Aged 15 years or above<br>• Resident in the catchment area<br>• Same gender as index case<br>• Age within 5 years of index case | • Past or current psychotic disorder<br>• Moderate or severe learning disability<br>• Clinically manifest organic cerebral disorder |
| **Relatives** | |
| • Aged 15 years or above<br>• Relative or carer for a person with psychosis recruited into the study | • Insufficient contact with person with psychosis to provide information on family burden and mental health |

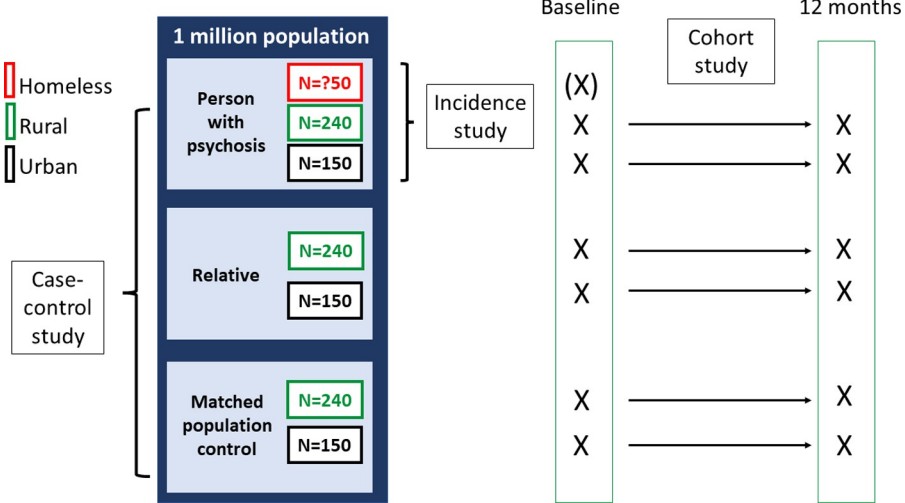

**Fig 2. Samples for epidemiological studies.**

**Sample size.** *Incidence.* With a conservative estimate of the population at risk (≥15 years) of 741,936 (rural population of 713,123 with an estimated 62% aged 15 years or above, and an urban population of 416,389 in 2016 with an estimated 72% aged 15 years or above) screened for incident cases over two years, assuming an incidence of psychosis of 18.7/100000 [88] person-years in the rural setting, this amounts to 139 new cases per year and approximately 554 people with psychosis including those who developed psychosis in the two years preceding baseline. Assuming 70% are identifiable and eligible, we expect to identify around 390 people with psychosis meeting our inclusion criteria. This sample size will have 80% power to detect an incidence rate ratio (IRR) of around 1.6 for urban exposure, in keeping with studies from high-income countries [89], and for males vs. females [88].

*Factors associated with onset of psychosis (case-control).* For a sample size of 390 people with psychosis and 390 controls, 16% early khat use in controls [39], we will have 80% power to detect an odds ratio of 1.7 for increased odds of early khat use in people with psychosis with p = 0.05.

*Follow-up study.* Assuming 20% loss to follow-up, we will have 80% power (p = 0.05) to detect a risk difference of 18% in receipt of minimally adequate care (medication prescription and attending for 4 follow-up appointments over 12 months [93]) assuming 30% receipt of minimally adequate care in non-khat chewers [94].

**Case-finding.** Following INTREPID methodology, building on our previous work [79] and arising from community engagement meetings, we will seek to identify people with potential psychosis through a combination of (1) proactive community-based case identification for those not in contact with care, and (2) case identification along help-seeking pathways. Proactive community-based case identification will comprise training of community health extension workers who carry out house-to-house visits for the 1000 households in their geographical catchment area, as well as other community informants, to identify people with possible psychosis. This approach has been shown to be sensitive for identification of people with psychosis in previous work by the Ethiopia team [29,95]. We will map out places where people with psychosis may seek help, including: health sector (specialist mental health services, general/primary healthcare services, health extension workers, private sector clinics); traditional and faith healing (churches, mosques, holy water sites, other healers); other sectors (police stations, prisons). We will identify focal persons (balanced by gender) in these sites and

provide them with orientation and illustrated written materials in local languages. We will ask them to record contact details for any persons with features of psychosis who present to them and to inform the study team. The research team will receive referrals from focal persons and contact them proactively on a monthly basis to identify new presentations of people with possible psychosis. Descriptions of psychosis for both community-based identification and identification on help-seeking pathways will be informed by existing knowledge of local manifestations, extensive stakeholder consultation, including people with lived experience of psychosis, and emerging findings from the epidemiological study [92]. Data collectors will also regularly review health facility contact data supplemented by a registry implemented by the research team to identify people with psychosis.

Our approach to early case identification will be adapted for identification of people who are homeless and have psychosis, mapping out places where people who are homeless commonly congregate, liaising with community police and making contact early in the morning to facilitate privacy [92]. To further facilitate identification, we will also carry out awareness-raising and evidence-based stigma reduction activities in local communities, in collaboration with people with psychosis, and expanding local accessibility of mental health care through integration in primary health care using effective models for this setting [72,94].

**Recruitment.** *Person with psychosis and relative.* Through linkage with the focal person/ key informant, SCOPE data collectors will seek to contact any individuals identified as having probable untreated psychosis with onset in the past two years. For non-homeless persons, the location of households will be identified by working with community-based health extension workers. The person with probable psychosis and a family member will be provided with information about the study. The baseline assessment will then be scheduled. At the baseline assessment, informed consent will be sought. Our approach to assessing capacity and seeking consent builds on previous work [74] and is described below under "key ethical considerations".

Once recruited, a mental health professional (psychiatrist or master's level mental health practitioner) will administer a screening questionnaire, including the Psychosis Screening Questionnaire (PSQ), and combine this with clinical observation and relative reports to assess likely eligibility for the main study [96]. The PSQ has been widely used in epidemiological studies to identify persons with probable psychosis. If there is any possibility of psychosis at this stage, the mental health professional will then administer a semi-structured diagnostic interview, the Schedules for Clinical Assessment in Neuropsychiatry (SCAN) [97], to confirm eligibility. A SCAN master trainer from the UK has trained Ethiopian mental health professionals in SCAN. Inter-rater and test-retest reliability will be evaluated. Previous studies have shown that SCAN-trained mental health professionals in Ethiopia have excellent reliability [13].

The full battery of baseline measures (see below) will then be administered to the individual with psychosis and the relative. The assessment will be split over two (or more) sessions to reduce respondent burden.

*Controls.* A control respondent will be identified by mapping the ten nearest neighbouring households and listing all adult residents by gender and age. All potential controls for the case (same sex, age within 5 years) will be approached in random order until an eligible control provides informed consent. This process will be repeated if no consenting control participant is found. The Psychosis Screening Questionnaire will be used to exclude current or past psychotic illness [96].

*People with psychosis who are homeless.* For people who are homeless with probable psychosis, we will find a private place for interview. Consent procedures are described below. People who are homeless and have probable psychosis who can participate in PSQ and SCAN will do so. For those who are too unwell, we will use methodology from our previous study of people

who are homeless and have psychosis and base diagnoses on systematic observation using SCAN [92].

All people with psychosis who are identified, regardless of study participation, will be referred to mental health care integrated within local primary health care (PHC) centres or psychiatric care if indicated.

**Assessment time-points and measures.**   See Table 2 for measures and assessment time-points. In the PRIME-Ethiopia evaluation of task-shared care for people with psychosis, a 12-month follow-up period was sufficient to model impact of baseline factors on patterns of engagement with care and clinical and social functioning outcomes [7]. Measures denoted with an asterisk* have been adapted and/or validated for the Ethiopian context previously. Laboratory investigations for malaria (where relevant), anaemia and tuberculosis seek to identify inequities in health in people with psychosis compared to controls. All are public health priorities within the Ethiopian context.

**Data collection and management.**   Mental health professionals trained in SCAN will conduct the diagnostic and semi-structured clinical assessments. Inter-rater reliability will be assessed periodically throughout the study. Lay data collectors with a minimum of completed secondary school education will administer the fully structured instruments. General health workers will collect the laboratory and TB investigations and assess physical health parameters. After training, we will select data collectors who have achieved proficiency. Based on previously developed procedures, we will ensure close supervision and data checking in the field, with weekly data quality checks and queries resolved alongside ongoing data collection. Data collection will be paper based for some aspects of the semi-structured measures, but mostly electronic (using smartphones or tablets) for structured interviews and stored securely on a server at Addis Ababa University. Upon completion of the main analyses, and after Ethiopian researchers have had full opportunity to make use of the data, de-identified datasets will be deposited with Addis Ababa University and made available for other researchers to access.

**Data analysis.**   We will use standard summary statistics, with indicators of spread and precision as appropriate (e.g., crude incidence rates per 100,000 person years, with 95% confidence intervals) to describe the data. We will then use appropriate regression models to compare data between rural and urban settings (e.g., Poisson regression for incidence rates and other count data; Cox regression for time-to-event data; logistic regression (including multinomial) for categorical data (e.g., course type); and linear regression for continuous data (e.g., General Assessment of Functioning score)). In building regression models, we will first fit univariable models, then test for effect modification by pre-specified variables (e.g., gender, age, setting and time) and finally adjust for potential confounders of each hypothesised association by fitting multivariable models.

## Aim 3: Co-development of interventions

The development of innovations to improve early identification and optimise recovery for people with psychosis in rural, urban and homeless populations will follow the Medical Research Council framework for development and evaluation of complex interventions [119]. This process will be informed by: (1) repeated Theory of Change workshops [120] with key stakeholders to harness local knowledge and (2) emerging findings from the epidemiological study, nested qualitative study, and community advisory group meetings, (3) the global evidence base through focused reviews, and (4) co-production workshops.

**Theory of Change (ToC) workshops.**   We will conduct repeated ToC workshops with key stakeholders in rural and urban sites (each n = 25) at months 8, 30 and 59. Stakeholders will include people with psychosis, caregivers, primary healthcare workers and managers, district

**Table 2. Measures and measurement timepoints.**

| Measure | Aim | Baseline | | | 12-month follow-up | | |
|---|---|---|---|---|---|---|---|
| | | Case | Relative | Control | Case | Relative | Control |
| Schedules for Assessment in Neuropsychiatry [97]* | 2a | X | | | | | |
| MRC socio-demographic schedule | 2a, 2b, 2c | X | X | X | X | X | X |
| Psychiatric and Personal History Schedule [17,98]* | 2a, 2c | X | X | | X | X | |
| WHO Life Chart Schedule [74,99]* | 2c | | | | X | X | |
| WHO Disability Assessment Schedule [100,101]* | 2a, 2c | X | X | X | X | X | X |
| Positive and Negative Symptom Scale (PANSS) [102]* | 2a, 2c | X | | | X | | |
| Patient Health Questionnaire– 9-item (depressive symptoms) validated for Ethiopia [103] | 2a, 2c | X | X | X | X | X | X |
| Generalised Anxiety Disorder-7 (GAD-7)* [104] | 2a, 2c | X | | | X | | |
| Global Assessment of Functioning [105]* | 2a, 2c | X | | | X | | |
| Pre-morbid adjustment scale [106] | 2a, 2b, 2c | X | X | | | | |
| List of threatening events [107]* | 2b, 2c | X | | X | X | | X |
| Alcohol, Smoking and Substance Involvement Screening Tool (ASSIST) [108]* | 2a, 2c | X | X | X | X | X | X |
| Problematic Khat Use Screening Tool* | 2a, 2b, 2c | X | X | X | X | X | X |
| Childhood trauma questionnaire [109] | 2b, 2c | X | | X | | | |
| Life Events Checklist for DSM-5 and PTSD Checklist for DSM-5 [110] * | 2a, 2c | X | | X | X | | X |
| Non-graphic screener for intimate partner violence [111] | 2a, 2c | X | | X | X | | X |
| Oslo Social Support Scale*[112] | 2a, 2c | X | | X | X | | X |
| Medication checklist [74] | | X | | X | X | | X |
| Glasgow Antipsychotic Side-effect Scale [113,114] | 2a, 2c | X | | | X | | |
| WHO-STEPS*[114] | 2a, 2c | X | | X | X | | X |
| Family Burden Interview Schedule*[115] | 2a, 2c | | X | | | X | |
| Laboratory investigation of malaria, anaemia | 2a, 2c | X | | X | X | | X |
| Body Mass Index | 2a, 2c | X | | X | X | | X |
| Tuberculosis screen | 2a, 2c | X | | X | X | | X |
| Adverse events [74] (mortality, homelessness, hospitalisation, victimisation, imprisonment, violence | 2c | X | | | X | | X |
| Household Food Insecurity Access Scale [116]* (HFIAS) | 2a, 2c | X | X | X | X | X | X |
| Camberwell Assessment of Need Short Appraisal Schedule [117]* (CANSAS) | 2a, 2c | X | | X | X | | X |
| Discrimination and stigma sub-scale on 'unfair treatment'*[118] (DISC-12) | 2a, 2c | X | | | X | | |
| NEW: Personal Recovery measure validated for setting | 2a, 2c | X | | | X | | |
| NEW: Family communication (scale to be selected and adapted) | 2a, 2c | X | | X | X | | X |

health administrators, mental health professionals, religious leaders, traditional healers, community representatives, representatives from social welfare, education and law enforcement; non-governmental organisation representatives, and policy-makers. The Ethiopia research team has extensive experience with ToC methodology, including mechanisms to ensure that marginalised voices are heard [121,122]. Separate ToC roadmaps will be identified for rural, urban and homeless populations. Differing needs for younger and older people with psychosis will be considered. At 30 months, revised ToC maps will identify priority components of contextually relevant models for early identification and intervention for untreated/first-contact psychosis in rural, urban and homeless populations, inform development of a programme theory, and inform the intervention evaluation (process evaluation and outcomes). Following feasibility testing of interventions, final ToC maps will be produced for each population.

**Co-production workshops.** Outputs from the ToC workshop will inform selection of the necessary components of the new interventions and implementation strategies for rural, urban and homeless populations, building on the global evidence base and local experience. People

with psychosis will join with small working groups of researchers and practitioners to co-produce interventions, using participatory methods used previously in Ethiopia [123,124].

## Aim 4: Interrupted time series study to evaluate identification strategies

We will investigate the impact of SCOPE interventions on identification and linkage of people with untreated psychosis to mental health care using an interrupted time series study design.

We have established recording forms for new cases of psychosis in 65 health facilities across the study site, including information on place of residence, age and gender. We will collect monthly data from the recording forms for the following time periods (1) pre-intervention (7–12 months), during intervention (13–36 months) and post-intervention (37–48 months). We will use segmented regression analysis [125] to examine change in the number of new cases of untreated psychosis contacting mental health care in the study areas, as has been used for previous similar studies. This will allow us to identify changes beyond those related to seasonal effects and secular trends.

## Aim 5: Feasibility trial and pilot studies of interventions

Protocols will be developed for a feasibility randomised controlled trial (RCT) and two uncontrolled pilot studies of the co-developed interventions. The feasibility RCT will address the research question: How feasible is a randomised controlled trial (RCT) of interventions for people with psychosis at first contact with services in rural districts in Ethiopia; what are implementation processes and outcomes? The uncontrolled pilot studies will use mixed methods pilot studies to investigate acceptability, feasibility, potential sustainability and perceived impact of innovations for people with psychosis homeless and/or living in urban settings. Outcomes measures will be determined from the Theory of Change maps, epidemiological study findings and priorities of people with lived experience of psychosis. Detailed protocols for these studies will be published separately.

## Key ethical considerations

Mental health services will be established or strengthened in all primary healthcare settings in the study recruitment sites so that persons identified via community case-finding procedures will have access to care locally. To ensure sustainability and integration into the existing health system, we are working closely with community advisory board members, including health administrators, to ensure necessary system supports are in place, including revolving drug funds to access psychotropic medications, training of psychiatric nurses to provide supervision and refresher training to primary healthcare workers, and ensuring that people with lived experience of mental health care are involved in planning and developing services. Specifically for people who are homeless and have SMI, we will follow similar procedures to our previous study on a homeless population in Ethiopia [92] and recommendations from our Community Advisory Boards, seeking to link people to mental health care services and social care resources that have been mapped within the study site.

Capacity to consent to participate in SCOPE studies will be evaluated by a mental health professional using a systematic and tested approach used previously in our research in Ethiopia [74]. If there is any uncertainty about whether the person has decision-making capacity, the person will be reviewed by a senior psychiatrist who is employed by the study team but not a member of the research team. If the individual lacks capacity to consent, a relative will be asked to provide permission for the individual to be included in the study. The justification for this approach is the ethical principle of justice, to ensure that services are informed by the needs of those who are most unwell, balanced against the risk to autonomy which we will

minimise by not including people who refuse and re-reviewing capacity. If the person subsequently regains capacity, we will then seek informed consent. If the person refuses participation at that stage, their data will be discarded. We have used this approach previously in Ethiopia [74].

For people who are homeless and have psychosis and who lack capacity to consent, we will identify a designated trusted individual who is well-placed to act in the person's interests, based on their preference, or, if they are unable to communicate a preference, a professional who is known to them (e.g. community health worker, religious leader). The person with psychosis and designated person will be informed about the study, and the designated person required to give permission. With our advisory board we will further develop methods for recruiting those who lack capacity and are unknown to community members. We will apply similar safeguards to the persons with psychosis who are not homeless but who lack capacity.

People with psychosis are at increased risk of harms and threats to their human rights in both community and healing settings (biomedical and traditional/faith healing), including restraint, physical or sexual harm, exploitation (financial, sexual, labour), neglect of physical and nutritional health; and unmet basic needs. We have worked closely with the community advisory boards and people with lived experience to develop rights-based safeguarding procedures that are contextually appropriate, to avoid unintended harms.

## Research uptake

We will seek to maximise research uptake through close working with our Advisory Board members and the Federal Ministry of Health of Ethiopia. In addition to publication in peer-reviewed journals, accessible policy briefs and summaries of key project findings will be prepared in the main languages in Ethiopia and used to engage with communities, as well as national and international organisations.

## Current status

The SCOPE project started formative phase activities on 22nd May 2022 and has completed the ethnographic study, scale development, piloting and validation study, and establishing surveillance methods for early case identification. We anticipate starting the epidemiological study in November 2023.

## Discussion

Findings from the SCOPE project will provide contextually grounded evidence from Ethiopia on the incidence of psychosis, aetiology of psychosis, unmet needs at first presentation and predictors of outcome that could be amenable to intervention. SCOPE will contribute to the small, but growing, number of longitudinal studies of people with psychosis across diverse LMICs. Through use of INTREPID methods, SCOPE data can be used in cross-country comparative analyses to illuminate our understanding of the aetiology and manifestations of psychoses and variations in the types of interventions required. SCOPE will also contribute rich, contextual understanding of the experiences of families affected by psychosis, leading to new measures of family communication and personal recovery that are relevant for this setting. Anchored in strong community engagement and involvement of people with lived experience, evidence generated by SCOPE will feed into co-development of interventions for earlier and better care that optimises recovery for people with psychosis. We will separately publish a detailed protocol to evaluate feasibility of a future trial, and the acceptability and feasibility of interventions.

The nature of interventions and associated implementation strategies will be informed by emerging data; however, the new interventions for people with psychosis at first contact with services will be located in primary health care rather than specialist mental health services, or sub-specialty early intervention for psychosis services. In part this decision reflects the extremely low population coverage of specialist mental health services in Ethiopia and other LMICs, which contributes to the current inaccessibility of mental health care, long duration of untreated psychosis and high out-of-pocket costs that impoverish families [32]. However, this decision is also informed by the need to promote parity of mental health and physical health care in primary health care and recognition of the benefits to the person with psychosis of staying close to community linkages to address social and economic needs.

Candidate interventions must be rights-based and community-focused, in line with best practice guidance from the World Health Organization [126]. Interventions may draw on critical time intervention (CTI) principles to develop proactive, phased and time-limited approaches to engaging people with psychosis in care to meet the multi-dimensional needs of individuals and their families [127] and/or may draw on elements of community-based rehabilitation with established effectiveness for people with chronic psychosis in rural Ethiopia [128]. Technological innovations have been used to support care for people with psychosis in high-income countries [129] and LMICs [130], although only in specialist settings. mHealth has the potential to link facility-based care with community-based health workers who can provide outreach, information and family support based on our previous materials [131,132] and facilitate ongoing engagement [130]. Globally, evidence for the contribution of peer support workers is also accumulating [133], and this approach would build on our previous work [121] and ensure relevance and sustainability. Integrated, task-shared approaches to delivering evidence-based brief psychosocial interventions [134–137] also hold promise in LMICs, including Ethiopia, with potential to address traumatic stress symptoms, substance use problems, depressive symptoms, adherence (including family involvement to support adherence [138]) and family support. Innovative decision support tools (technology-guided) and care planning to promote physical health and address under-nutrition may have applicability in mortality reduction.

Interventions for people with psychosis who are homeless will build on previous experience with training community health workers and police for awareness-raising, case-identification and supportive outreach of people who are homeless and have severe mental illness [92]. We will consider innovative ways of addressing unmet physical health, mental health and social needs, while overcoming access barriers. While provision of housing is considered an essential first step to treatment in high-income countries [139], this is not feasible in a low-income country setting, and many do not fulfil criteria for psychiatric admission [92]. Alternative approaches need to be considered to initiate treatment while a person is still homeless. Given the complexity of the SCOPE study, we anticipate that there may need to be modifications made to the protocol to ensure feasibility and achieve the aims. These will be documented, justified and included in publications of the SCOPE findings.

## Conclusions

The SCOPE project seeks to contribute high quality evidence on the sociocultural context of psychosis in Ethiopia and create momentum for earlier and better care through the development and testing of feasible, acceptable and scalable interventions to increase earlier uptake of care and at first contact with services to optimise recovery.

## Supporting information

**S1 File. Detailed description of settings.**
(DOCX)

**S2 File. Details of formative research.**
(DOCX)

## Acknowledgments

CH, AA and EM receive support from the National Institute for Health and Care Research (NIHR) through the NIHR Global Health Research Group on Homelessness and Mental Health in Africa (NIHR134325) and CH also receives support from the SPARK project (NIHR200842) using UK aid from the UK Government. The views expressed in this publication are those of the authors and not necessarily those of the NIHR or the Department of Health and Social Care. CH also receives support from WT grant 223615/Z/21/Z. TR receives a fellowship from the British Academy (PF21\210001). CL receives support from the National Institute for Health Research (NIHR) (using the UK's Official Development Assistance (ODA) Funding) and Wellcome (grant number: 221940/Z/20/Z) under the Department of Health and Social Care (DHSC)-Wellcome Partnership for Global Health Research. CM is part-funded by the ESRC (ESRC Centre for Society and Mental Health at King's College London: ESRC Reference: ES/S012567/1). For the purposes of open access, the author has applied a Creative Commons Attribution (CC BY) licence to any Accepted Author Manuscript version arising from this submission.

## Author Contributions

**Conceptualization:** Charlotte Hanlon, Tessa Roberts, Craig Morgan.

**Data curation:** Mekonnen Tsehay, Dessalegn Kiross.

**Funding acquisition:** Charlotte Hanlon, Tessa Roberts, Eleni Misganaw, Ashok Malla, Alex Cohen, Crick Lund, Abebaw Fekadu, Craig Morgan, Atalay Alem.

**Methodology:** Charlotte Hanlon, Tessa Roberts, Eleni Misganaw, Ashok Malla, Alex Cohen, Teshome Shibre, Wubalem Fekadu, Solomon Teferra, Derege Kebede, Adiyam Mulushoa, Mekonnen Tsehay, Dessalegn Kiross, Crick Lund, Abebaw Fekadu, Craig Morgan, Atalay Alem.

**Project administration:** Charlotte Hanlon, Adiyam Mulushoa, Zerihun Girma.

**Supervision:** Charlotte Hanlon.

**Validation:** Wubalem Fekadu.

**Writing – original draft:** Charlotte Hanlon.

**Writing – review & editing:** Tessa Roberts, Eleni Misganaw, Ashok Malla, Alex Cohen, Teshome Shibre, Wubalem Fekadu, Solomon Teferra, Derege Kebede, Adiyam Mulushoa, Zerihun Girma, Mekonnen Tsehay, Dessalegn Kiross, Crick Lund, Abebaw Fekadu, Craig Morgan, Atalay Alem.

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
