## [Decision Letter · Decision Letter 0]

11 Jan 2024

PONE-D-23-32426Studying the context of psychoses to improve outcomes in Ethiopia (SCOPE): protocol paperPLOS ONE

Dear Dr. Hanlon,

Thank you for submitting your manuscript to PLOS ONE. After careful consideration, we feel that it requires minor revision in order to be suitable for publication. Therefore, we invite you to submit a revised version of the manuscript that addresses the points raised during the review process.

We look forward to receiving your revised manuscript.

Kind regards,

Annika C. Sweetland, DrPH, MSW

Academic Editor

PLOS ONE

Reviewers' comments:

Reviewer's Responses to Questions

**Comments to the Author**

1. Does the manuscript provide a valid rationale for the proposed study, with clearly identified and justified research questions?

Reviewer #1: Yes

Reviewer #2: Yes

2. Is the protocol technically sound and planned in a manner that will lead to a meaningful outcome and allow testing the stated hypotheses?

Reviewer #1: Yes

Reviewer #2: Yes

3. Is the methodology feasible and described in sufficient detail to allow the work to be replicable?

Reviewer #1: Yes

Reviewer #2: Yes

4. Have the authors described where all data underlying the findings will be made available when the study is complete?

Reviewer #1: No

Reviewer #2: Yes

5. Is the manuscript presented in an intelligible fashion and written in standard English?

Reviewer #1: Yes

Reviewer #2: Yes

6. Review Comments to the Author

You may also provide optional suggestions and comments to authors that they might find helpful in planning their study.

Reviewer #1: First, I would like to congratulate the research team, the protocol is well written and detailed for possible replication but I would like to leave some comments:

1. In Background’s protocol, it would be important to address the multifactorial etiology of schizophrenia or psychoses, genetic and socio-environmental, to not leave the impression that psychoses or schizophrenia has only socio-environmental etiology.

2. In lines 249 and 251, bibliographies that reinforce or justify the statistical numbers presented in the protocol must be included.

3. In line 264 of the protocol, which talks about adapting situation analysis tools used in the PRIME study, in the ethical considerations is not mentioned that the request for permission to use these tools was done.

4. In line 334 about the eligibility criteria used in INTREPID, it is not clear in the protocol whether or not permission was requested, to use data from this study, or whether they are for general use.

5. In line 267, about ethnographic studies, it is not specified how many families will participate in this qualitative study and what the sampling process will be. I think it would be important to be included in the main document. It should also be specified how many researchers will be involved in this ethnographic study and what contingency measures will be taken in the event of a psychotic crisis in the patient in the family under study.

6. In line 448, in the table 2, it would be better to clarify why the patient will have to test for anemia, malaria and tuberculosis.

7. In relation to homeless people who are identified as having psychosis, it must be clear in the protocol how these patients will be monitored after the end of the study.

8. In the downloaded protocol, the figures do not appear, only the titles.

9. It is not clear how the results of the study will be made available.

Reviewer #2: Reviewer’s comments for Studying the context of psychoses to improve outcomes in Ethiopia (SCOPE): protocol paper for PLOS ONE

This programme of work is an important addition to several others being carried out in the Global South aimed at understanding the nature of severe mental disorders and meeting the needs of those affected by these conditions within the cultural and social context of where they reside. It builds on the rich tradition of psychosis research in Ethiopia and extends the ongoing efforts in that country to expand services for persons living with psychosis. It has the potential of providing a model on which other countries across the developing world can pattern their aspirations of increasing access to nuanced mental health services to numerous individuals affected by psychosis and their families in the different settings. Below are some areas in the manuscript requiring clarification:

1. Eligibility criteria for SCOPE epidemiological study: the authors wish to reduce the lower limit of eligibility age to 15 years. It is important that the investigators provide some reason/ explanation for this decision to extend the lower age limit of their sample to 15 years and comment on the ethical implications of this decision, and how they wish to address it, in the manuscript.

2. For the purpose of SCOPE, how do the investigators define “homelessness”? The concept of homelessness can mean different things depending on whether it is being discussed in within Western or African cultural contexts, and may also have urban-rural differences in meaning. It is important to include this definition in the manuscript, even if briefly.

3. Page 18, line 420. I suspect you mean “test-retest”, not “test-restest”.

4. Page 21, line 460-462: “Data collection will be paper based for some aspects of the semi-structured measures, but mostly electronic (using smartphones or tablets) for structured interviews and stored securely on the university server.” Which university are the authors referring to?

5. Ethical considerations (page 23). While the authors have provided extensive information on the process of obtaining consent on the field, it does happen from time to time that research staff on the field are unsure whether some potential respondents have capacity to consent or not. Could the authors comment on how this will be resolved if and when it occurs? Will senior investigators be contacted to resolve uncertainties in the capacity of potential respondents to give consent? Will another mental health practitioner unconnected with the study be employed to determine capacity to consent if the authors wish to be mindful of conflict of interest?

6. Formative qualitative study: Regarding the ethnography component of the study, entire households will be observed by the investigators. How do the authors wish to approach consent? Will this be from the head of the household or each member of the household? What happens some members of the household do not give consent for ethnographic study, when the head of the household has?

7. PLOS authors have the option to publish the peer review history of their article (what does this mean?). If published, this will include your full peer review and any attached files.

Reviewer #1: **Yes: **Rogério João Mulumba

Reviewer #2: No

---

## [Author Response · Author response to Decision Letter 0]

16 Feb 2024

Correspondence address:

Department of Psychiatry

7th Floor, College of Health Sciences Building

Tikur Anbessa Hospital compound

PO 9086

Addis Ababa, Ethiopia

Phone: +251 966 253760

14th February 2024

Dr Emily Chenette

Editor-in-chief

PLoS One Journal

Dear Dr Chenette

Re: Response to reviewers 

PONE-D-23-32426

 Studying the context of psychoses to improve outcomes in Ethiopia (SCOPE): protocol paper

Thank you for your invitation to revise our manuscript. We appreciate the thoughtful feedback from the reviewers. Please see below for a point-by-point response.

Reviewer #1: 

First, I would like to congratulate the research team, the protocol is well written and detailed for possible replication but I would like to leave some comments.

Response: Thank you for your kind and helpful comments.

1. In Background’s protocol, it would be important to address the multifactorial etiology of schizophrenia or psychoses, genetic and socio-environmental, to not leave the impression that psychoses or schizophrenia has only socio-environmental etiology.

Response 1

We have now added the following sentence to the background:

The aetiology of psychosis is multifactorial, contributed to by social and environmental risk factors alongside genetic and developmental risks [1].

2. In lines 249 and 251, bibliographies that reinforce or justify the statistical numbers presented in the protocol must be included.

Response 2

We have added the references, as follows:

The studies will be conducted in two sites: (1) contiguous, predominantly rural districts in south-central Ethiopia (Gurage zone of Southern Nations, Nationalities and People’s (SNNP) Region: Misrak Meskan, Merab Meskan, Sodo and South Sodo; Oromia region: Sodo Daci and Kersana Malima; Special district: Sabata Hawas) with an estimated total population of 713,123 people [2]; and (2) Lideta and Kirkos sub-cities of Addis Ababa, the capital of Ethiopia, with an estimated total population of 416,389 in 2016 [3].

3. In line 264 of the protocol, which talks about adapting situation analysis tools used in the PRIME study, in the ethical considerations is not mentioned that the request for permission to use these tools was done.

Response 3

We have clarified that study co-author (CH) led development of the PRIME situation analysis tool. The cited reference supports this. No permissions are required to use or adapt the PRIME situation analysis tool.

4. In line 334 about the eligibility criteria used in INTREPID, it is not clear in the protocol whether or not permission was requested, to use data from this study, or whether they are for general use.

Response 4

We have clarified that the INTREPID eligibility criteria are published in a protocol. No permission is required to use the INTREPID methods. However, we note that three of our co-investigators (CM, AC, TR), all co-authors on this paper, are INTREPID investigators. 

5. In line 267, about ethnographic studies, it is not specified how many families will participate in this qualitative study and what the sampling process will be. I think it would be important to be included in the main document. It should also be specified how many researchers will be involved in this ethnographic study and what contingency measures will be taken in the event of a psychotic crisis in the patient in the family under study.

Response 5

We have added the following into the main manuscript:

Ethnographic observations in 12-20 households of people with psychosis are combined with 20-30 in-depth interviews with a range of stakeholders (people with psychosis, caregivers, mental health providers, community leaders). These will investigate patterns of family interaction, impacts of mental ill-health and the status of the individual with psychosis in the family. 

Families will be purposively selected based on urban/rural location, trajectory of illness, and educational level of household head, recruited from the Butajira psychiatric clinic or Sodo district mental health care services (rural site), or Lideta sub-city health centres or mental health services (Addis Ababa). The person with psychosis will be required to provide informed consent and all members of the household should agree to the observation. The household head will also provide informed consent. A researcher will spend two hours at a time with each family, scheduled for different times of the day, to observe family members’ activities on arrival and their interactions with the person with psychosis. Each household with a person with psychosis will be observed for an estimated 30 to 40 hours over a period of six months. The observations will be conducted by master’s level research assistants, one male and one female depending on the gender of the person with psychosis. If a person with psychosis becomes unwell and requires mental health care, the researcher will liaise with a senior mental health professional in the team and advise the family to support the person to access mental health care.

6. In line 448, in the table 2, it would be better to clarify why the patient will have to test for anaemia, malaria and tuberculosis.

Response 6

We have added the following justification for these tests:

Laboratory investigations for malaria (where relevant), anaemia and tuberculosis seek to identify inequities in health in people with psychosis compared to controls. All are public health priorities within the Ethiopian context.

7. In relation to homeless people who are identified as having psychosis, it must be clear in the protocol how these patients will be monitored after the end of the study.

Response 7

We will not monitor people after the end of the study, which is cross-sectional in nature. However, we have added the following text to describe our efforts to support people who are homeless and have psychosis:

Specifically for people who are homeless and have SMI, we will follow similar procedures to our previous study on a homeless population in Ethiopia [4] and recommendations from our Community Advisory Board, seeking to link people to mental health care services and social care resources that have been mapped within the study site. 

8. In the downloaded protocol, the figures do not appear, only the titles.

Response 8

We apologise for the inconvenience. On the PLOS ONE platform, they must be submitted separately. I hope you will be able to access them to download separate from the main manuscript. 

9. It is not clear how the results of the study will be made available.

Response 9

We have added the following text to the ‘data collection and management section’

Upon completion of the main analyses, and after Ethiopian researchers have had full opportunity to make use of the data, de-identified datasets will be deposited with Addis Ababa University and made available for other researchers to access.

We have also added a section on ‘research uptake’, as follows:

Research uptake

We will seek to maximise research uptake through close working with our Advisory Board members and the Federal Ministry of Health of Ethiopia. In addition to publication in peer-reviewed journals, accessible policy briefs and summaries of key project findings will be prepared in the main languages in Ethiopia and used to engage with communities, as well as national and international organisations. 

Reviewer #2: Reviewer’s comments for Studying the context of psychoses to improve outcomes in Ethiopia (SCOPE): protocol paper for PLOS ONE

This programme of work is an important addition to several others being carried out in the Global South aimed at understanding the nature of severe mental disorders and meeting the needs of those affected by these conditions within the cultural and social context of where they reside. It builds on the rich tradition of psychosis research in Ethiopia and extends the ongoing efforts in that country to expand services for persons living with psychosis. It has the potential of providing a model on which other countries across the developing world can pattern their aspirations of increasing access to nuanced mental health services to numerous individuals affected by psychosis and their families in the different settings. 

Thank you for your kind comments and constructive feedback.

10. Eligibility criteria for SCOPE epidemiological study: the authors wish to reduce the lower limit of eligibility age to 15 years. It is important that the investigators provide some reason/ explanation for this decision to extend the lower age limit of their sample to 15 years and comment on the ethical implications of this decision, and how they wish to address it, in the manuscript.

Response 10

We have added the following justification and comment on the ethical implications and how these will be addressed:

The reason for lowering the age for inclusion is because many cases of psychosis start in late adolescence [5]. We will seek informed consent from those aged under 18 years. If they are an emancipated minor, we will not seek permission from a guardian/responsible family member. If they are aged 15-17 years and not an emancipated minor, we will seek permission from that person before including in the study.

11. For the purpose of SCOPE, how do the investigators define “homelessness”? The concept of homelessness can mean different things depending on whether it is being discussed in within Western or African cultural contexts, and may also have urban-rural differences in meaning. It is important to include this definition in the manuscript, even if briefly.

Response 11

We have added the following text to describe our working definition of homelessness, 

Our operationalisation of the concept of homelessness was informed by stakeholders in the community advisory board. Homelessness is thus defined here as spending the night unsheltered or in other places not intended for habitation (e.g. under bridges). It includes people who can sporadically pay for shelter but excludes people who spend their days on the streets, for example, to beg, but who have stable night-time accommodation. 

12. Page 18, line 420. I suspect you mean “test-retest”, not “test-restest”.

Response 12: Yes! Thank you for spotting that. 

13. Page 21, line 460-462: “Data collection will be paper based for some aspects of the semi-structured measures, but mostly electronic (using smartphones or tablets) for structured interviews and stored securely on the university server.” Which university are the authors referring to?

Response 13

We have clarified that this is Addis Ababa University.

14. Ethical considerations (page 23). While the authors have provided extensive information on the process of obtaining consent on the field, it does happen from time to time that research staff on the field are unsure whether some potential respondents have capacity to consent or not. Could the authors comment on how this will be resolved if and when it occurs? Will senior investigators be contacted to resolve uncertainties in the capacity of potential respondents to give consent? Will another mental health practitioner unconnected with the study be employed to determine capacity to consent if the authors wish to be mindful of conflict of interest?

Response 14

We have added further detail as per the reviewer’s recommendations:

If there is any uncertainty about whether the person has decision-making capacity, the person will be reviewed by a senior psychiatrist who is employed by the study team but not a member of the research team.

15. Formative qualitative study: Regarding the ethnography component of the study, entire households will be observed by the investigators. How do the authors wish to approach consent? Will this be from the head of the household or each member of the household? What happens some members of the household do not give consent for ethnographic study, when the head of the household has?

Response 15

Please see Response 5 to reviewer 1 where we cover these issues.

Yours sincerely

Charlotte Hanlon

1. Owen MJ, Sawa A, Mortensen PB. Schizophrenia. The Lancet. 2016;388(10039):86-97. doi: 10.1016/S0140-6736(15)01121-6.

2. Central Statistical Agency. Population Projections for Ethiopia: 2007-2037. Addis Ababa, Ethiopia: CSA, 2013.

3. Tsehay M, Shibre Kelkile T, Fekadu W, Cohen A, Misganaw E, Hanlon C. Mapping resources available for early identification and recovery-oriented intervention for people with psychosis in Addis Ababa, Ethiopia. medRxiv. 2024:2024.01.16.24301385. doi: 10.1101/2024.01.16.24301385.

4. Fekadu A, Hanlon C, Gebre-Eyesus E, Agedew M, Haddis S, Teferra S, et al. Burden of mental disorders and unmet needs among street homeless people in Addis Ababa, Ethiopia. BMC medicine. 2014;12(138):20.08.2014.

5. Häfner H, Löffler W, Maurer K, Riecher-Rössler A. The Influence of Age and Sex on the Onset and Early Course of Schizophrenia. British Journal of Psychiatry. 1993;162(1):80-6. Epub 2018/01/03. doi: 10.1192/bjp.162.1.80.

---

## [Decision Letter · Decision Letter 1]

11 Apr 2024

Studying the context of psychoses to improve outcomes in Ethiopia (SCOPE): protocol paper

PONE-D-23-32426R1

Dear Dr. Hanlon,

We’re pleased to inform you that your manuscript has been judged scientifically suitable for publication and will be formally accepted for publication once it meets all outstanding technical requirements.

Kind regards,

Annika C. Sweetland, DrPH, MSW

Academic Editor

PLOS ONE

Reviewers' comments:

Reviewer's Responses to Questions

**Comments to the Author**

1. Does the manuscript provide a valid rationale for the proposed study, with clearly identified and justified research questions?

Reviewer #1: Yes

Reviewer #2: Yes

2. Is the protocol technically sound and planned in a manner that will lead to a meaningful outcome and allow testing the stated hypotheses?

Reviewer #1: Yes

Reviewer #2: Yes

3. Is the methodology feasible and described in sufficient detail to allow the work to be replicable?

Reviewer #1: Yes

Reviewer #2: Yes

4. Have the authors described where all data underlying the findings will be made available when the study is complete?

Reviewer #1: Yes

Reviewer #2: Yes

5. Is the manuscript presented in an intelligible fashion and written in standard English?

Reviewer #1: Yes

Reviewer #2: Yes

6. Review Comments to the Author

You may also provide optional suggestions and comments to authors that they might find helpful in planning their study.

Reviewer #1: In my opinion, the new version is ready to be published. I think the study will greatly help people who suffer from psychosis, who are often neglected.

Reviewer #2: Dear Editor,

I have reviewed the response of the authors and their revision of the manuscript and found them to be satisfactory.

I recommend accepting the manuscript for publication.

Regards.

Olatunde O Ayinde.

7. PLOS authors have the option to publish the peer review history of their article (what does this mean?). If published, this will include your full peer review and any attached files.

Reviewer #1: **Yes: **Rogério Mulumba

Reviewer #2: **Yes: **Olatunde O Ayinde

---

## [Editor Report · Acceptance letter]

26 Apr 2024

PONE-D-23-32426R1 

PLOS ONE

Dear Dr. Hanlon, 

I'm pleased to inform you that your manuscript has been deemed suitable for publication in PLOS ONE. Congratulations! Your manuscript is now being handed over to our production team.

Kind regards, 

on behalf of

Dr. Annika C. Sweetland 

Academic Editor

PLOS ONE